# Improving Cancer Diagnosis in Alberta, Canada: A Qualitative Study of Emergency Department Healthcare Providers’ Perspectives on Diagnosing Cancer in the Emergency Setting

**DOI:** 10.3390/curroncol32010005

**Published:** 2024-12-25

**Authors:** Anna Pujadas Botey, Cassandra Carrier, Eddy Lang, Paula J. Robson

**Affiliations:** 1Cancer Research & Analytics, Cancer Care Alberta, Alberta Health Services, Calgary, AB T2N 2T9, Canada; 2School of Public Health, University of Alberta, Edmonton, AB T6G 1C9, Canada; paula.robson@ahs.ca; 3Emergency Department, South Health Campus, Alberta Health Services, Calgary, AB T3M 1M4, Canada; cassandra.carrier@ahs.ca; 4Department of Emergency Medicine, Cumming School of Medicine, University of Calgary, Rockyview General Hospital, Calgary, AB T2V 1P9, Canada; eddy.lang@ahs.ca; 5Cancer Research & Analytics, Cancer Care Alberta, Alberta Health Services, Edmonton, AB T5J 3H1, Canada

**Keywords:** cancer diagnosis, cancer care, diagnostic process, emergency department, qualitative research, healthcare providers’ perspectives, care coordination

## Abstract

Cancer is the leading cause of death in Canada, with diagnoses increasing annually. In Alberta, many cancer cases are detected in emergency departments, often at advanced stages. Despite the significant role of emergency departments in cancer diagnosis, limited research exists on the experiences of healthcare providers in this context. This qualitative study aimed to explore the perspectives of physicians and nurses working in emergency departments in Edmonton and Calgary regarding cancer diagnosis. Semi-structured interviews were conducted with 17 physicians and nurses, recruited through convenience and snowball sampling. Data collection continued until thematic saturation was reached. Interviews were analyzed thematically using an inductive, iterative process. Three main themes emerged: the acute care focus of the emergency department, its unsuitability for cancer diagnosis, and the need for systemic improvements to better support patients with suspected cancer. Participants highlighted challenges related to high patient volumes, the emotional burden of delivering cancer diagnoses, and barriers to effective communication and patient interaction in a fast-paced, high-pressure environment. The findings suggest the need for systemic reforms, including stronger primary care and improved care coordination, to alleviate pressure on emergency departments and enhance both patient outcomes and healthcare provider well-being.

## 1. Introduction

Cancer is the leading cause of death in Canada [1]. Following national trends, it is estimated that one in two Albertans will be diagnosed with cancer in their lifetime, and at least one in five will die of the disease [2]. From 2000 to 2019, the one-year prevalence of all cancer in the province has increased by 85% (from 9627 to 17,810 [3]). This upward trend is expected to continue, with cancer incidence projected to rise by 23% by 2030 and by 36% by 2040, driven by population growth and aging [3].

Administrative data show that up to 35–40% of patients diagnosed with the most common cancers in Alberta are diagnosed after visiting an emergency department (ED) (Cancer SCN, internal data, 2016–2021). Similar findings have been reported in Ontario, where 35% of patients visited the ED within 90 days of their cancer diagnosis [4], and globally, with studies highlighting comparable rates in jurisdictions such as the United Kingdom and other high-income countries [5,6,7]. These patients are often diagnosed with later-stage cancers, which are more challenging to treat and are associated with lower quality of life and reduced 5-year survival rates relative to patients diagnosed at earlier stages [8,9,10,11,12]. Many patients delay seeking care due to factors to such as fear of diagnosis, symptom ambiguity, or limited awareness of the severity of their condition, which often leads to acute symptom escalation and ED visits [8,13,14]. This underscores the critical role of EDs play as an entry point for cancer diagnosis, particularly for patients who experience acute symptom escalation or lack regular access to primary care [15,16,17]. Factors contributing to this lack of access include shortages of primary care providers, particularly in rural and remote areas; geographic barriers, which disproportionately impact rural and remote populations; and socioeconomic challenges such as financial constraints, lower health literacy, and stigma associated with certain diagnostic procedures or conditions [17,18,19]. Together, these systemic and individual factors highlight the complexity of ED reliance for cancer diagnosis and its implications for patient outcomes and healthcare systems.

For patients diagnosed after visiting an ED, the implications can be significant. These diagnoses have been associated with increased psychological distress, as patients often face the dual burden of processing a life-changing diagnosis while managing acute symptoms that prompted their visit [20,21]. The sudden and unexpected nature of receiving a cancer diagnosis in the ED can exacerbate feelings of vulnerability, anxiety, and loss of control for both patients and their families [20,22]. The ED’s fast-paced and impersonal environment may further amplify these challenges, limiting opportunities for meaningful patient–provider communication and emotional support [22,23,24]. These experiences can have lasting effects on patients’ psychological well-being and their ability to cope with the demands of cancer treatment.

Despite the recognition of EDs as crucial in cancer diagnosis, the existing literature predominantly focuses on characterizing patients visiting EDs [4,25,26,27] and determining their outcomes and epidemiological trends such as the prevalence of late-stage diagnoses and associated mortality rates [16,28,29,30]. There is a lack of qualitative data exploring the experiences and perspectives of ED healthcare providers, who are on the front lines of this diagnostic pathway. Previous studies have acknowledged the significant role ED healthcare providers play in identifying and diagnosing cancer [26,31]. However, there is a paucity of research investigating how these providers perceive their role and responsibilities and the unique challenges they face in the ED context. There is limited understanding of how to better support ED clinical teams in managing these cases, which often involve not only medical but also emotional and ethical challenges.

This gap in the literature points to a need for a deeper exploration of the ED teams’ experiences and insights, which can inform targeted interventions to improve patient care and outcomes. Given this background, the objective of this study is to explore the perspectives and experiences of a group of healthcare providers (physicians, nurses) working in EDs in the two major cities in Alberta—Edmonton and Calgary—regarding cancer diagnosis in the emergency setting. The study aims to elucidate the perceived roles and responsibilities of the ED teams, identify the needs and challenges they encounter, and gather their recommendations for enhancing support for patients with suspected cancer. By addressing these underexplored areas, the study seeks to generate valuable knowledge that can contribute to better clinical practices and the implementation of initiatives like the Alberta Cancer Diagnosis Initiative [32,33], ultimately aiming to improve patient outcomes and experiences across Alberta. The insights gained are anticipated to have broader implications for improving cancer diagnosis and patient care in emergency settings beyond Alberta, providing a foundation for similar research and intervention strategies in diverse healthcare contexts.

## 2. Materials and Methods

### 2.1. Design

A qualitative study was performed utilizing interviews guided by an interpretive description methodology [34]. This approach enabled us to examine the diagnostic process through the lens of ED healthcare providers, uncovering detailed themes and patterns in their personal experiences and perspectives.

### 2.2. Setting and Participants

Convenience sampling [35] was used to recruit physicians and nurses working in EDs in Edmonton and Calgary, drawing upon existing connections of the Emergency Strategic Clinical Network and the emergency nursing team at South Health Campus, Calgary. These individuals were invited to participate in the project through emails, meeting discussions, and advertisements in social media forums. In addition, snowball sampling [35] was employed, whereby participants were asked to recommend physician or nurse colleagues who might be interested in participating. Eligible participants were physicians and nurses working in adult-only EDs in Edmonton or Calgary who were interested and in a position to discuss cancer diagnosis. The profile of respondents was tracked as they made contact with the research team, and it became necessary to deny participation to just one individual who worked in a pediatric ED.

### 2.3. Data Collection

Semi-structured, in-depth interviews were carried out. The interview guide was designed by two authors (APC and CC), drawing on insights from the existing literature and prior studies exploring the perspectives of both patients and healthcare providers [13,31]. Interview topics are presented in Table 1. The interview guide was pilot-tested with three staff members in an ED, and small adjustments were made to improve clarity and flow. Before commencing their interviews, participants verbally confirmed their informed consent.

The interviews were conducted by one author (CC), an ED registered nurse and Quality Improvement Lead for an ED in Calgary. CC has extensive experience supporting ED patients and interacting with ED personnel, and had prior professional relationships or interactions with some participants. She has keen interest and some experience in research and was involved in the study from its inception. Interviews were conducted via phone or Microsoft Teams. No non-participants were present during the interviews, and no repeat interviews were conducted. All participants completed the study without dropping out. During each interview, field notes were taken to capture contextual details. Interviews took place between October 2023 and March 2024.

### 2.4. Data Analysis

Participants were added until data saturation was achieved, indicating that no new themes emerged during the analysis of subsequent interviews [36]. All interviews were recorded, transcribed verbatim, and processed using NVivo Version 12 (QSR International, Burlington, MA, USA). A thematic analysis [34] was conducted using an inductive, data-driven coding process to understand how participants interpreted their experiences [37]. This process involved carefully reviewing each transcript, identifying preliminary themes, and iteratively refining them as data collection and analysis progressed. One author (APB), in collaboration with the interviewer (CC), organized themes into codes that were applied to text fragments in the transcripts. To ensure consistency and trustworthiness [37], a second researcher coded randomly selected segments, with both researchers discussing their interpretations and codes until reaching consensus.

## 3. Results

Seventeen participants were interviewed, including eight physicians and nine nurses. Participants worked at seven different hospitals: four in Edmonton and three in Calgary. Over one-third of participants (35%, n = 6) had worked in an ED for 10 to 14 years, and 41% (n = 7) had worked there for 9 years or less. The majority were women (71%, n = 12), and the median age of participants was 41 years (range 25–71). Interviews lasted 44 min on average (range 27–57).

Thematic analysis revealed three salient themes as being relevant to the experiences and perspectives of participants: (1) ED teams focus on acute presentations, (2) the ED is not the ideal environment for cancer diagnosis, and (3) improved ED cancer diagnosis requires reducing patient volumes and optimizing diagnosis processes across the healthcare system. Appendix A presents themes and subthemes.

### 3.1. ED Teams Focus on Acute Presentations

Participants consistently described the primary role of ED teams as addressing acute medical conditions and providing immediate care. One participant explained, “*in ED you’re going from a problem-focused approach*” (P10), while another emphasized, “*the main focus is on symptom management and prevention of disease progression*” (P1). Participants noted that ED teams handle acute issues rather than providing ongoing, comprehensive care, differentiating their responsibilities from those of primary care providers. As one participant expressed, “*Emergency is only for emergencies, right? [our role is] to stabilize them, to treat, and to manage and to work up and investigate. But it isn’t often full patient health; it’s only very specific complaints that are managed*” (P1). Another clarified, “*[we’re] treating their symptoms, their pain, or their nausea, whatever they might have happening when they come in*” (P8).

A secondary role, often referred to as either by choice or out of necessity, involves the initial diagnosis of conditions that may present as incidental findings, such as cancer. One participant noted, “*it can be quite tough in the emergency setting to find something incidentally and not act on that and just act on what their main presentation is*” (P1). Others shared similar experiences: “*I would work up the symptom as much as possible, imaging and blood work, until diagnosis if possible*” (P17), and “*I think in general, from a responsibility standpoint, people [patients] expect that if people come in with symptoms related to cancer […], we should either be making the diagnosis or at least thinking about the possibility of the diagnosis*” (P16). Participants also noted how patients come in for certain acute symptoms and in some cases unexpectedly receive critical news about cancer, “*I think one of the big things is that often it’s an unexpected finding… We are often breaking bad news and having that discussion of what we found on imaging or testing […]. That is often a challenge for a patient to come in and say, ‘Well, I have a sore stomach. What do you mean it’s cancer?’.*” (P3).

Beyond initial diagnoses, ED teams ensure that patients have the next steps in care, which are typically provided outside the ED or acute care settings. This involves coordinating with other healthcare services for definitive diagnosis, follow-up referrals and patient treatment. One participant shared, “*If no diagnosis is made in the ED, we often do let the patient know this is a concerning symptom, and make sure they do get follow up*” (P17). When a diagnosis is made, “*[Our] role is in kind of getting people that we have caught set up to start treatment and start connecting with all the right people that they need to connect with to kind of further their treatment going forward*” (P6). One participant elaborated: “*We typically may not have these patients with us for some time. If it’s a bowel obstruction, secondary to cancer, then they’re going to come into the building. Often our patients will get diagnosed and go home and then they’ll have their follow up with the Tom Baker [cancer centre] or a specialist down the road. I find we’re more in the diagnostic phase. We’re there telling them, we’ve found the cancer and letting them know […], and sending on consultations for follow-up.*” (P8).

Participants also discussed the evolving responsibilities of ED healthcare providers, especially in recent years. Many remarked on the increased pressure on ED physicians to perform tasks traditionally handled by primary care physicians. One participant observed, “*That’s all family doctor kind of stuff, but by default, since people don’t have family doctors, we’re the last line of defense so we’ve got to do it*” (P15). This was further clarified by others who noted, “*a lot of times people don’t have routine access to their GPs or even have a GP*” (P4), or “*[they] don’t have or can’t access [a/their GP] in a timely manner or haven’t seen them*” (P5). Another participant explained, “*family doctors are dealing with huge constraints and a crazy amount of administrative burden that’s making their jobs untenable*” (P16). These shifts have significantly expanded the role of ED physicians: “*Right now, as opposed to three years ago, I -as the emergency doc- have to do a lot of heavy lifting in terms of next steps for the patient. Whereas previously I referred them back to their family doctor to do things like refer or book a biopsy, refer to the cancer centre, ask questions about prognosis, [and] prescribe symptom management, […]. So, three years ago, the process was: give the diagnosis, provide some supportive care, and then send [patients] back to the family doctor […]. And now, […] I try to do as much for them as I can in terms of getting biopsies organized and leading with cancer centres and oncologists. And I spend a lot of extra time doing that.*” (P5) “*I think ideally, we would be the people who would send the referral on for someone else to quarterback the diagnosis and management. […] In real life, the people that we see diagnosed don’t have anyone else, and that’s why they come through the emerge. In the end we end up being the quarterback, often for months, until they’re taken over by the Cross Cancer [cancer centre]. From diagnosis to biopsy to referrals, we do it all.*” *(P17).*

### 3.2. The ED Is Not the Ideal Environment for Cancer Diagnosis

Participants consistently described the ED as a challenging environment for diagnosing cancer and delivering such significant news to patients. They referred to the ED as “*a horrible place to get told you have cancer*” (P10) and “*an unfortunate place for it to happen*” (P12). The environment was often explained as ill-suited for providing compassionate care, with one participant stating, “*the ED environment is not the most conducive to receiving life-changing news*” (P6).

Several barriers were identified that make the ED suboptimal for sensitive conversations related to cancer diagnosis. The first barrier is the lack of privacy, and the inherently busy nature of the ED. Participants highlighted how the curtained rooms and crowded spaces contribute to a stressful environment. One participant described a distressing scenario where patients receive a cancer diagnosis in a crowded waiting room, “*You’re sitting in a waiting room with 40 other people, and a doctor bends down and tells you that you have cancer—that’s horrible*” (P10). The lack of appropriate space for patients to receive and process their diagnosis was further emphasized: “*The majority of people are learning something life-changing and traumatizing in an area where they’re not given the space and ability to process that information. […]. Getting bad news in the ED is hard enough. Learning that you have cancer when you have zero privacy to process that, that’s unacceptable.*” (P10).

The high-pressure environment of the ED often forces healthcare providers into a task-oriented approach, which can impede the ability to spend adequate time with patients. Participants noted the impact of that on patient care, stating, “*There’s a ton going on and [providers] are on a very tight timeframe of how long they actually get to spend with each patient. There’s always competing priorities*” (P12). Another added, “*It is so busy, we end up getting caught up with our tasks and having not to great therapeutic relationship with the patients. Sometimes it just ends up being task, task, task, ‘Are you stable? OK, let’s move on’.*” (P1).

Participants also highlighted the lack of information available to healthcare providers when delivering a cancer diagnosis in the ED. This limitation not only affects the immediate conversation but also leaves patients uncertain about what will happen next, adding to their distress. One participant noted that patients often have questions about the extent of their illness and the steps that will follow, to which healthcare providers frequently have no answers, “*A lot of times they have questions that you just don’t have the answers to. You know, ‘How far has this spread?’ ‘What does this mean?’ is a big one, ‘What’s next?’ ‘How bad is it?’*” (P7). This uncertainty is further exacerbated by the often-vague follow-up process after leaving the ED, as described by another participant: “*If I see somebody in the ED, I’m like, ‘I think you almost certainly have cancer,’ and then give them the news. Then I’m like, ‘You should get a phone call next week.’ I have no idea what is the time course. I’m usually just guessing. I don’t really know […]. I think that is so shitty for patients to be left with, ‘I don’t know if I have cancer or not. I think the doctor told me I have cancer.’ People ask some questions, and it’s just so lousy.*” *(P16)*.

The emotional difficulty of receiving a cancer diagnosis is amplified in the ED because patients are typically unprepared for such news. Participants reflected on the shock that patients experience, with one noting, “*Usually, they’ve had some sort of symptom […]. I’ll say they had abdominal pain and expect to be told they are constipated or something, not being told they have cancer*” (P12). Another participant expressed the emotional toll on both patients and healthcare providers: “*I diagnose and break the bad news to the patient about their cancer diagnosis. That’s usually a shock to them and sometimes a shock to me too, and really sucks*” (P5). Participants further emphasized the emotional challenges encountered by healthcare providers, particularly when they must convey an uncertain but probable diagnosis that will later be confirmed by specialist care outside the ED: “*That puts us in a tough position […]. How do we communicate these results […]? Not wanting to panic them or be wrong […] That’s a hugely impactful thing to say to somebody […]. I think it’s very tricky*” (P14).

Finally, the lack of an established patient–provider relationship in the ED further complicates the delivery of a cancer diagnosis. One participant observed, “*We don’t really know these people; we haven’t had time. I think it’s very, very tricky*” (P14). Another noted, “*They meet someone for the first time who tells them about their cancer. They don’t know me. I don’t know them*” (P5). This challenge is compounded by the lack of continuity in care, as patients often see different healthcare providers during the diagnostic process. One participant illustrated this by saying, “*I might be the first physician that saw them, and I’ll send them for a CT scan. They return and see someone completely different. That person reviewing those results hasn’t had any contact with that patient at all*” (P3). Participants suggested that cancer diagnosis conversations would be better handled by someone with whom the patient has an ongoing relationship, such as a family physician. As one participant expressed, “*If it was me, I’d rather be having that conversation with a family doctor that I’ve met many times before and I have an established relationship with*” (P16).

### 3.3. Improved ED Cancer Diagnosis Requires Reducing Patient Volumes and Optimizing Diagnosis Processes Across the Healthcare System

Participants were asked about potential improvements to cancer diagnosis within the ED. Their responses primarily focused on strategies to reduce the volume of incoming patients with suspected cancer and to optimize cancer diagnostic processes outside of the ED. Rather than suggesting direct changes to the ED itself, participants emphasized the need to alleviate the ED’s burden by addressing upstream factors.

Participants highlighted the importance of health promotion and public awareness as crucial measures to reduce patient load in the ED. They advocated for expanding public health campaigns to educate individuals about prevention and early detection of cancer. One participant suggested practical tools like “*patient handouts […], having a website*” that could guide people on “*changes you can do in your diet, changes you can do in your lifestyle, [and] things to look out for*” (P1). Another stressed the value of community-level education initiatives, such as “*reminding men during Movember to check their testicles. Those kind of community education bits [...] are really valuable*” (P10). Enhanced public education on cancer screening guidelines was also seen as essential for facilitating early detection and reducing reliance on emergency care: “*Some better guidelines that are more readily available and out there in the public*” (P12).

Participants underscored the critical role of improving primary care, suggesting that strengthening primary care could reduce the number of patients turning to the ED for cancer-related issues. One participant noted, “*Bolster primary care so it does what it’s supposed to do. For the occasional patient that slips through the cracks, we’ll see them in emergency*” (P15). Others raised concerns about suboptimal access to primary care and stressed the need for “*more GPs in the province [...], greater accessibility to primary care and to actually getting reasonable testing done in the community in a reasonable time frame*” (P10) and “*better access to imaging, biopsies, and cancer clinics to ensure prompt diagnosis*” (P17). Improved patient access to primary care was also described as vital for preventing avoidable ED visits: “*If people had an opportunity to see their family doctor within a day or two instead of five to six weeks... would be a great place to start*” (P7).

Participants identified the creation of a centralized intake, triage, and referral system specifically for cancer diagnosis as an ideal solution to address the challenges in both primary and emergency care. This system would coordinate the diagnostic process, ensuring timely and appropriate care while preventing patients from getting lost in the system and ending in an ED. Participants envisioned a comprehensive support network that would include resources like “*a help line [...], a physician assistant line [...], a nurse assistant line [...], [and] a general public assistance line*” (P1) to facilitate a more efficient and seamless care process. This system would enable “*a more streamlined referral process, minimizing wait times and ensuring that patients don’t get lost in the system*” (P1), and would help alleviate some of the burden on ED teams, allowing them to focus on critical care while the system handled coordination and follow-up. As one participant noted, “*I can do my CT and say, this looks like cancer and I’m referring you to somebody who [...] is going to support the questions about [...] disability insurance [and] all of the social aspects*” (P5). A dedicated medical professional, particularly a nurse with expertise in cancer care, was seen as crucial to the success of this system. They would serve as the main point of contact for patients and healthcare providers, answering questions, explaining the care process, and triaging patients by assessing which symptoms require urgent attention versus those that can be managed in primary care or through self-management. By providing necessary interventions, such as pain management, for patients who do not require immediate ED care, this professional would help bridge gaps in the care continuum. They would also offer guidance to patients and coordinate with primary care physicians. As one participant explained, “*If your pain medication is working, I can either give advice to your family doctor who’s managing it, or if you don’t have one, I can manage it until you get seen*” (P17).

## 4. Discussion

This study contributes to the literature by focusing on the perspectives of ED healthcare providers regarding cancer diagnosis. The findings highlight the crucial role ED teams play in identifying cancer, while also revealing the significant challenges they face in fulfilling this role. Participants’ narratives highlight that these challenges are exacerbated by a systemic issue leading to a heavy reliance on EDs for cancer diagnosis.

Our findings show that healthcare providers in the ED often assume responsibilities beyond what might be their formal roles, acting to fill critical gaps in patient care. Despite the lack of clarity in their roles, participants expressed a strong sense of duty, driven by the belief that they are the only ones who have the access and capacity to perform these essential tasks. This aligns with the existing literature, where ED healthcare providers have reported feeling responsible for patient care, particularly in crisis situations like the COVID-19 pandemic, driven by an intrinsic professional obligation rather than external expectations [38]. Similarly, the authors in [39] emphasize that ED healthcare providers view their professional duty as a personal commitment, independent of legal or ethical mandates.

As discussed in the literature and supported by our findings, this commitment to patient care reflects not only a strong sense of responsibility, but also resilience among ED staff. They adapt to the chaotic ED environment by leveraging their experience and attributes such as efficiency, creativity, and the ability to manage high-pressure situations [38]. However, relying on informal role-taking might introduce risks. While our findings do not explicitly discuss variability in care delivery, the absence of formalized roles and clear responsibilities can undermine the consistency of patient care, a concern echoed in other studies [40,41]. Additionally, as other studies highlight, these unclear roles and informal practices can cause frustration among healthcare providers, who must balance patient needs with overwhelming workloads [42]. This ongoing strain in high-pressure environments contributes to stress and burnout [38,42], underscoring the need to prioritize provider well-being and job satisfaction amid increasing pressures in EDs [42].

Our study further highlights how the chaotic nature of the ED hinders effective communication and patient interaction, also leading to lower patient care satisfaction levels—a problem previously identified in the literature [42,43]. Participants in our study reported significant barriers, such as limited time and a lack of privacy when delivering complex diagnoses like cancer, which increase patient distress and frustration for both patients and healthcare providers.

To address these challenges and improve patient care in the ED, our findings suggest that broader, systemic improvements, rather than isolated modifications within the ED itself, are needed. Participants stressed the importance of addressing upstream factors, such as strengthening primary care and enhancing early detection through public health campaigns, to reduce avoidable ED visits. Additionally, our findings support the need for streamlining care through better coordination and the development of a centralized referral system. Aligning with the goals of the Alberta Cancer Diagnosis Initiative [32,33], these improvements could alleviate pressures on the ED and enhance both patient and provider experiences. This comprehensive approach is crucial as poor care coordination continues to exacerbate the growing demands on EDs [42], and will support not only cancer diagnosis in EDs but across all healthcare settings. Future research could focus on evaluating the implementation and impact of these systemic improvements, conducting longitudinal studies to assess long-term outcomes, exploring stakeholder perspectives on these improvements, and analyzing their cost-effectiveness. Comparative studies across different healthcare systems could also provide valuable insights into best practices and effective strategies for improving cancer diagnosis and care coordination.

### Limitations

Despite the valuable insights gained from this study, several limitations should be noted. First, the use of convenience and snowball sampling, while effective for reaching participants within the ED network—particularly ED healthcare providers in Edmonton and Calgary—may introduce selection bias and may not fully represent the diverse experiences of these teams. Second, the involvement of our interviewer as an ED nurse with prior professional relationships with some participants could introduce bias, despite efforts to maintain neutrality. Third, due to time constraints, transcripts were not returned to participants for comment or correction, missing an opportunity to deepen the validation of our findings through participant feedback. Finally, the study was conducted within a specific timeframe and geographical context (EDs in Edmonton and Calgary), which may affect the applicability of the findings to other times or locations. Despite these limitations, results from the study make significant contributions to understating ED cancer diagnosis from the perspective of healthcare providers. They also suggest avenues for future research, emphasizing the need for a more diverse participant pool from different geographic locations and ED types to enhance the generalizability of the findings. Additionally, incorporating longitudinal designs and multiple data collection methods, such as observations and surveys, could provide a more comprehensive understanding of the challenges and practices related to cancer diagnosis in emergency settings. Engaging participants in the validation process through member checking could further strengthen the reliability of the findings.

## 5. Conclusions

This study underscores the vital role of ED healthcare providers in cancer diagnosis and reveals significant challenges they face due to systemic issues and informal role-taking. The findings highlight the necessity for comprehensive, systemic improvements to address upstream factors like primary care and public health initiatives. Streamlining care through better coordination and a centralized referral system can alleviate pressures on EDs and improve patient and provider experiences. By addressing these issues, we can enhance cancer diagnosis and patient care across all healthcare settings, ultimately leading to more effective and compassionate care for patients with cancer.

## Figures and Tables

**Table 1 curroncol-32-00005-t001:** Topics included in interview guide.

Responsibilities of ED teams in cancer diagnosis
Collaboration between ED teams and other healthcare providers
Steps after symptom presentation in the ED
Needs and challenges in diagnosing cancer in the ED
Obstacles hindering early cancer diagnosis
Suggestions for improving cancer diagnosis processes (in ED and beyond)

## Data Availability

The datasets generated and/or analyzed during the current study are not publicly available due confidentiality, but excerpts are available from the corresponding author on reasonable request.

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
