# Peer review of "Improving Cancer Diagnosis in Alberta, Canada: A Qualitative Study of Emergency Department Healthcare Providers’ Perspectives on Diagnosing Cancer in the Emergency Setting"

_curroncol, 2024, doi:10.3390/curroncol32010005_

Round 1

Reviewer 1 Report

Comments and Suggestions for Authors

The authors conducted a qualitative study on healthcare providers’ perspectives regarding cancer diagnosis at an emergency department (ED). The study is well conducted and presented. The authors reported the content in a clear, synthetic, and organized manner. I have only some suggestions:

Introduction

The rationale, study novelty, knowledge gap, and thesis statement are well formulated. However, I suggest the authors expand the first paragraph of the introduction on cancer epidemiology. The second paragraph will start with “Administrative data show that up to 35-40% of patients diagnosed with the most common cancers in Alberta are diagnosed after visiting an emergency department (ED) (Cancer SCN, internal 42 data, 2016-2021)…….”. Also this second paragraph should be expanded. In addition, I suggest the authors add a paragraph on patients’ implications (this paragraph will be the third) of having a cancer diagnosis at an ED.

Results

The results are presented clearly. However, I suggest the authors be more specific in labeling the three main themes that emerged from the analysis. The authors have brought out considerations and implications (quite obvious) rather than detecting and labeling the main themes.

Furthermore, I suggest the authors add a table with themes and subthemes along with participants’ quotes.

Reviewer 2 Report

Comments and Suggestions for Authors

The authors present a very interesting article titled: “Improving cancer diagnosis in Alberta, Canada: A qualitative study of emergency department healthcare providers’ perspectives on diagnosing cancer in the emergency setting”. They examined the challenges of diagnosing cancer in the emergency department from the perspective of staff.

Comments and suggestions:

-        The authors state that 35-40% of cancers are diagnosed after a visit to an emergency department. It would be helpful to clarify whether this percentage is specific to Alberta, Canada, or if it reflects a global trend. Providing additional context on the underlying reasons for this high percentage would strengthen the introduction.

-        In line 48, the authors mention that the emergency department serves as an entry point for patients "who may not have access to primary care." It would be valuable to elaborate on the reasons contributing to this lack of access. For example, are there specific factors related to the local healthcare system, geographical barriers, or socioeconomic conditions that lead to this phenomenon?

-        The results section contains valuable information and is enriched with numerous quotes from study participants. The cited statements are highly relatable and provide meaningful insights.

-        I appreciate the authors' suggestions for future research and the comprehensive description of the study's limitations.

In summary, this manuscript is well-written and addresses a topic of high relevance to everyday clinical practice. The inclusion of numerous participant experiences offers valuable insights into the challenges practitioners face when diagnosing cancer in the emergency department.

Round 2

Reviewer 1 Report

Comments and Suggestions for Authors

The authors have addressed all my concerns. The table with themes and subthemes significantly improves the clarity of the results.